# Boosting Efficiency in Task-Agnostic Exploration Through Causal Knowledge

## Abstract

The effectiveness of model training heavily relies on the quality of available training resources. However, budget constraints often impose limitations on data collection efforts. To tackle this challenge, we introduce *causal exploration* in this paper, a strategy that leverages the underlying causal knowledge for both data collection and model training. We, in particular, focus on enhancing the sample efficiency and reliability of the world model learning within the domain of task-agnostic reinforcement learning. During the exploration phase, the agent actively selects actions expected to yield causal insights most beneficial for world model training. Concurrently, the causal knowledge is acquired and incrementally refined with the ongoing collection of data. We demonstrate that causal exploration aids in learning accurate world models using fewer data and provide theoretical guarantees for its convergence. Empirical experiments, on both synthetic data and real-world applications, further validate the benefits of causal exploration.

## 1 Introduction and related work

Deep neural network models have been incredibly successful in various domains, such as the milestone achievements in Go games and control tasks (Silver et al., 2016; Tassa et al., 2018). One key factor contributing to such remarkable performance is the availability of high-quality data for model training. However, in many practical applications, it remains data-hungry due to limited data collection efforts imposed by budget constraints (Settles, 2009; Fang et al., 2017; Yoo & Kweon, 2019; Robine et al., 2023).

To tackle this challenge for data collection and model training, we introduce causal exploration in this paper, a novel framework that makes use of the underlying causal knowledge to improve learning efficiency. Causal discovery, a fundamental process involving the identification of causal relationships between variables within a system, plays a crucial role in both processes (Pearl et al., 2000). Acquiring and understanding such causal knowledge unveils the fundamental mechanisms behind the data generation process, thus enhancing the performance of the learned model (Molina et al., 2020; Jaber et al., 2020; Peng et al., 2022).

Specifically, we focus on boosting the sample efficiency and reliability of the world model in the realm of task-agnostic reinforcement learning (RL). Different from methods that learn a fixed task from scratch, task-agnostic RL agent first learns a global model that gathers information about the true environment on the data collected during exploration. The learning process is exclusively driven by intrinsic rewards, which measure the agent's level of surprise at the outcome. Then based on the predictions of the learned world model, the agent can make quick adaptations to downstream tasks in a zero-shot manner provided with specific reward functions. However, the data collection and world model learning processes are usually expensive due to extensive interactions with the environment, especially in large state spaces where discovering the optimal policy can be highly challenging. Hence, we introduce causal exploration to efficiently learn world models with causal knowledge.

Our causal exploration-based approach revolves around three primary aspects. First, how to extract and identify causal knowledge of the environment? In this regard, we employ causal models to capture the factored Markov decision processes over state transitions and use constraint-based methods to discover causal relationships among environment variables.

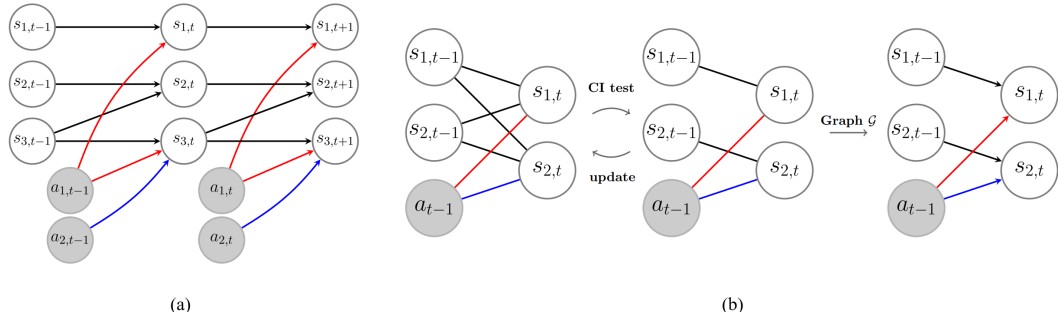

(a)                                                                    (b)

Figure 1: Illustrations of (a) causal relationships among variables in the RL system (b) causal discovery with conditional independence (CI) tests.

Second, how to effectively utilize causal structures to enhance the reliability of a learned world model? That is, when predicting future outcomes, the model is supposed to only take the parent nodes as inputs according to the causal graph. Prior methods like ASR (Huang et al., 2022) embed structural constraints into the world model and update them together through continuous optimization. However, the adoption of such embedding methods often results in increased model complexity and necessitates additional computational resources for training, which tends to yield sub-optimal solutions in data-hungry scenarios (Liang et al., 2019; Van Leeuwen et al., 2021; Zhang & Zhang, 2023). In light of these potential limitations, we decouple the discovery of causal structure from world model learning and design a sharing-decomposition schema to ensure that the model's initialization and updating processes adhere to the causal constraints by zeroing out non-parent nodes. Causal knowledge and the world model are continuously refined with the ongoing collection of data.

Third, how can we improve sample efficiency during the exploration process? While causal discovery algorithms typically necessitate the collection of substantial causal information through data, it's important to note that accumulating more samples does not always confer an advantage: as the sample size increases, the time cost of causal algorithms also rises. Hence, prioritizing the enhancement of data quality over quantity becomes paramount. In this paper, we present an efficient online causal discovery method, based on PC (Spirtes et al., 2000). Instead of indiscriminately accumulating all the coming data, our approach selectively eliminates redundant and noisy samples, strategically gathering informative data points for causal discovery in an incremental manner.

On the other hand, selecting representative samples is advantageous for reducing the cost of model training. We draw inspiration from active learning methods (Holub et al., 2008; Siddiqui et al., 2020) to improve the data collection efficiency. To be specific, the agent explores by selecting actions that will lead to samples with the largest contribution to the model training loss. The prediction error is then calculated with a scaling factor as the agent's intrinsic motivation. During causal exploration, the agent keeps searching in the state space to select data that the model mostly unfamiliar with and maximize its expected intrinsic rewards, while the world model optimizes to minimize the prediction loss. These two steps facilitate each other through continuous exploration.

The work most closely related to ours is that of Seitzer et al. (2021). However, their focus is primarily on detecting causal influences between actions and future states. In contrast, our approach extends to encompass causal relationships among states as well. Furthermore, our objective centers around enhancing exploration efficiency from an active learning standpoint, while the exploration policy in Seitzer et al. (2021) relies on random sampling, which is usually less effective in scenarios involving large state spaces. Our key contributions are summarized below.

- In order to enhance the sample efficiency and reliability of model training with causal knowledge, we introduce a novel concept: causal exploration, and focus particularly on the domain of task-agnostic reinforcement learning.
- To efficiently learn and use structural constraints during world model learning, we design a novel weight-sharing-decomposition schema that can avoid additional computational burden.
- We demonstrate the effectiveness of causal exploration across a range of demanding reinforcement learning environments. Theoretically, we show that, given strong convexity and smoothness assumptions, our approach attains a superior convergence rate compared to non-causal methods. Empirically, experimental results on synthetic data demonstrate the ability to learn accurate world models with exceptional data utilization, surpassing existing exploration methods in complex sce-

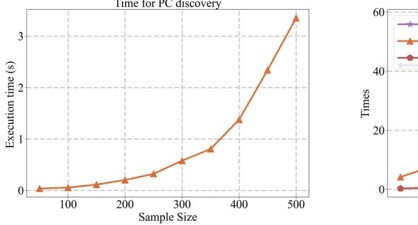 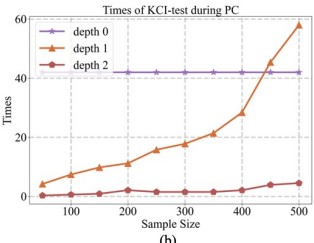 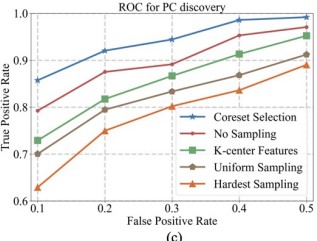

Figure 2: (a)(b) Computational cost for the PC algorithm with KCI-tests. (c) Performance of online version of PC using different coreset selection methods.

narios. Notably, our approach also produces outstanding performance in challenging tasks such as traffic light control and MuJoCo, highlighting its practicality in real-world applications.

# 2 DISCOVERING AND UTILIZING CAUSALITY FOR WORLD MODELS

We focus on causal exploration within a Markov decision process characterized by a factored state space $\mathcal{S} = \mathcal{S}_1 \times \ldots \times \mathcal{S}_n \in \mathbb{R}^n$ and action space $\mathcal{A} = \mathcal{A}_1 \times \ldots \times \mathcal{A}_c \in \mathbb{R}^c$. We take into account the causal structure over the state-action variables $\mathcal{V} = \{s_{1,t-1}, \ldots, s_{n,t-1}, a_{1,t-1}, \ldots, a_{c,t-1}, s_{1,t}, \ldots, s_{n,t}\}$, which can be represented by a binary matrix $D \in \{0, 1\}^{(n+c) \times n}$ indicating the structure from $(s_{t-1}, a_{t-1})$ to $s_t$. Take the example in Figure 1(a): we have $D(1, 1) = 1$ because $s_{1,t}$ is causally related to $s_{1,t-1}$, while $D(2, 1) = 0$ because $s_{1,t}$ does not have a causal edge to $s_{2,t-1}$. We assume that the structural constraints are invariant over time $t$. In the following part of this section, we first give the identification procedure of the causal graph, and then we show how to utilize the learned causal knowledge for model training.

## 2.1 EFFICIENT ONLINE CAUSAL RELATIONSHIP DISCOVERY

To identify the causal structure represented by the binary matrix $D$, a commonly employed approach is the PC algorithm, which leverages conditional independence(CI) constraints implied by the data. Roughly speaking, the original PC algorithm starts from a complete undirected graph $\mathcal{C}$ over all variables in the vertex set $\mathcal{V}$, and then removes the edge between each ordered pair of adjacent vertices by testing CI. Edges are oriented for triples of vertices by further employing the results from conditional independence tests, as well as the acyclic constraints. In our setting, the cardinality of the vertex set $\mathcal{V}$ is $2n + c$ which contains state-action sequences $\{s_t, a_t\}$ and next-state sequences $\{s_{t+1}\}$, and we only care about the structural relationships among state transitions. That is, the initial causal graph $\mathcal{G}$ is a subgraph of $\mathcal{C}$ where instantaneous edges between variables have already been dropped out as shown in Figure 1(b). Moreover, the causal directions can be directly oriented by leveraging the temporal order.

Note that the reliability of the PC algorithm indeed requires substantial data collection. However, the continual accumulation of data does not always result in favorable outcomes. Actually, the implementation of causal discovery often encounters an computational bottleneck as the number of samples increases. Figure 2(a) gives an example where the execution time of the PC algorithm based on Kernel-based Conditional(KCI) test (Zhang et al., 2012) experiences exponential growth. Therefore, in order to reduce the cost of the identification process, we design an efficient online causal relationship discovery method: rather than use all of the coming data for causal identification, we selectively collect representative data points during exploration in an incremental way. Specifically, we use the minibatch similarity and sample diversity criteria introduced in Yoon et al. (2021) as our selection strategies, which are defined as

$$\text{Similarity} = \frac{\nabla f_{\boldsymbol{w}}(b_t^i) \overline{\nabla} f_{\boldsymbol{w}}(B_t)^T}{\|\nabla f_{\boldsymbol{w}}(b_t^i)\| \cdot \|\overline{\nabla} f_{\boldsymbol{w}}(B_t)^T\|}, \quad \text{Diversity} = \frac{-1}{t-1} \sum_{p \neq i}^{t-1} \frac{\nabla f_{\boldsymbol{w}}(b_t^i) \overline{\nabla} f_{\boldsymbol{w}}(b_t^p)^T}{\|\nabla f_{\boldsymbol{w}}(b_t^i)\| \cdot \|\overline{\nabla} f_{\boldsymbol{w}}(b_t^p)^T\|}. \tag{1}$$

Here $b_t^i$ is the $i$-th data of the whole arrival batch $B_t$. $f$ is the world model in our method with parameter $\boldsymbol{w}$, $\nabla f_{\boldsymbol{w}}(b_t^i)$ and $\overline{\nabla} f_{\boldsymbol{w}}(B_t)$ are the gradient and average gradient of the sample and batch, respectively. A combination of batch similarity and diversity is used to rank and select the top-$k$

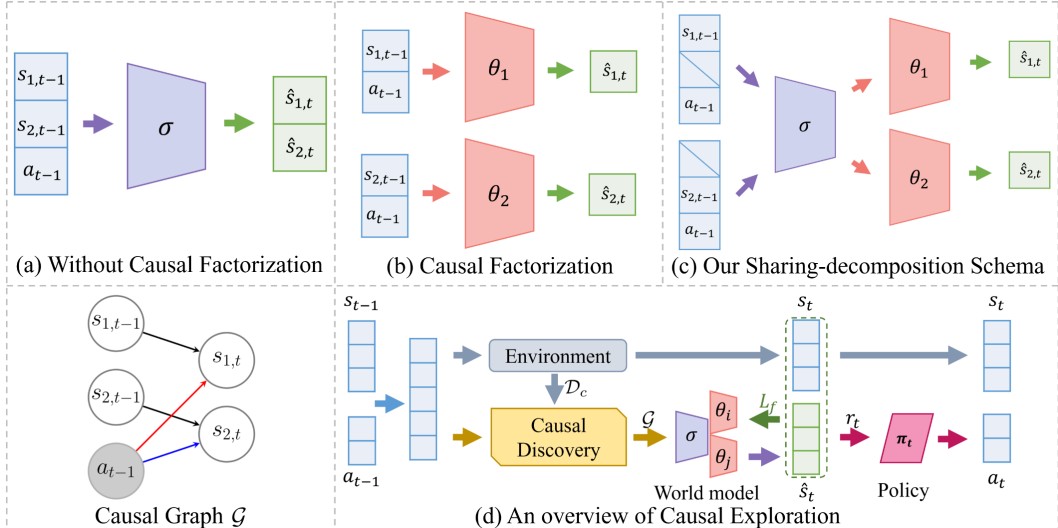

Figure 3: (a)(b)An illustration of model architecture before and after causal factorization; (c) An illustration of the sharing-decomposition schema; (d)An overview of Causal Exploration framework.

data points of $B_t$ for causal discovery during exploration. Experiments in section 4 have shown that this online method significantly accelerates causal discovery without sacrificing overall accuracy.

## 2.2 CAUSAL CONSTRAINTS FOR FORWARD MODEL

After acquiring the causal knowledge, we show how to make use of it for model training. Forward world model $f_{\boldsymbol{w}}$ enables agents to predict future state $\hat{\boldsymbol{s}}_{t+1}$ based on current state $\boldsymbol{s}_t$ and action $\boldsymbol{a}_t$, $f_{\boldsymbol{w}} : (\boldsymbol{s}_t, \boldsymbol{a}_t) \to \boldsymbol{s}_{t+1}$. Typically, the prediction loss is then used to optimize the network parameters $w$ as in Curiosity-driven Exploration (Burda et al., 2018a):

$$L_f = \frac{1}{2}\|f_{\boldsymbol{w}}(\boldsymbol{s}_t, \boldsymbol{a}_t) - \boldsymbol{s}_{t+1}\|_2^2, \tag{2}$$

where the neural network employs fully connected layers. However, Huang et al. (2022) shows that such a framework could be redundant because causal graphs are often not fully connected in the data generation process. For example, $s_{1,t+1}$ is only causally related to $s_{1,t}$ and $a_{1,t}$ but not connected with $s_{2,t}, s_{3,t}$ and $a_{2,t}$ as illustrated in Figure 1(a). Therefore, we are suggested to partially take $s_{1,t}$ together with $a_{1,t}$ as input instead of $(\boldsymbol{s}_t, \boldsymbol{a}_t)$. Utilizing this causal structure over variables to formulate and learn a world model is believed to be more reliable and efficient.

The challenge for leveraging causal information into a world model is that each dimension has its unique parents and needs special design. Instead of using a single model to predict next state $\hat{\boldsymbol{s}}_{t+1}$, we are supposed to design $n$ decomposed neural networks $f_{\boldsymbol{w}_i^c}$ for the prediction of each factored dimension $\hat{s}_{i,t+1}$, formulated as

$$f_{\boldsymbol{w}^c} = \prod_{i=1}^{n} f_{\boldsymbol{w}_i^c}(Pa_{\mathcal{G}}(s_{i,t+1})). \tag{3}$$

Figure 3(a) and (b) illustrate an example before and after factorization under the causal graph $\mathcal{G}$ in Figure 1(b), respectively. Such method would be sufficient for simple models like single layer neural network. However, it is likely to result in an explosive growth in computational complexity as the state dimension and network size increase. This goes against our prime intention.

We propose a sharing-decomposition schema to address such a problem. It is unnecessary for all of these $n$ networks to be totally different. In other words, each of these models could share the first several layers and design their specific architecture following the sharing module. Suppose $\boldsymbol{w}_i^c$ is the network parameter for the $i$-th dimension, it is a combination of the shared parameter $\boldsymbol{\sigma}$ and decomposed parameter $\boldsymbol{\theta}_i$, written as $\boldsymbol{w}_i^c = \boldsymbol{\sigma} \cup \boldsymbol{\theta}_i$. What's more, to reflect these causal constraints for each forward model, we zero out values of the state and action nodes that are not parents of the decomposed state (such as $s_{2,t}$ for $s_{1,t+1}$ in Figure 1(a)) in the input layer of $\boldsymbol{\sigma}$. During the training time, each decomposed model focuses on a different aspect of the state in the

decomposition part $\boldsymbol{\theta}_i$ but shares a common knowledge $\boldsymbol{\sigma}$. The number of the shared layers is a hyperparameter that allows for a trade-off between the sharing and decomposition parts. By training forward models under this schema, our approach can both utilize causal information of the ground environment dynamics to generate accurate predictions and achieve a significant reduction in model parameters and computation time compared with absolute decomposition. Figure 3(c) illustrates our sharing-decomposition schema during causal exploration.

## 3 BOOSTING EFFICIENCY THROUGH CAUSAL EXPLORATION

We now return to the fundamental question: how to enhance the data collection efficiency during causal exploration, thereby improving the performance of both causal discovery and model learning. The agent, following an exploration policy $\pi$, is supposed to take actions that are expected to continually acquire data most beneficial for model training and updating causal beliefs. To attain this goal, a commonly applied concept from active learning is the selection of samples that make the largest contributions to the model's training loss. These samples are typically considered as a subset that the model is least familiar with. Hence, the prediction loss is used here as the intrinsic reward to guide exploration with a scaling weight $\eta$:

$$r_t^i = \frac{\eta}{2}\|f(\boldsymbol{s}_t, \boldsymbol{a}_t) - \boldsymbol{s}_{t+1}\|_2^2. \tag{4}$$

This prediction loss can also be viewed as a validation of the agent's causal beliefs. The larger the prediction error, the more surprised the agent is by the actual outcome, implying a greater deviation from the estimated values based on the causal structure and the world model. The faster the error rate drops, the more learning progress signals we acquire.

Apart from the prediction loss, we introduce active reward (Fang et al., 2017) as another intrinsic motivation to guide agent to actively explore towards causal informative data. Note that not all data has a positive impact on the model. On the contrary, some redundant data may lead the model to an awful direction. To reflect this, we collect a test set $\mathcal{D}_h$ generated from episodes unseen before training. The prediction accuracy on the test set reflects world model's prediction ability. Once the world model is updated, we calculate the change of performance before/after update as active reward that provides feedback on the quality of training, indicating a beneficial/detrimental training caused by the selected data. If the reward is always positive, it indicates that the agent has been selecting beneficial samples for training the world model. We formulate active reward as

$$r_t^a = e(\phi_{t-1}) - e(\phi_t), \tag{5}$$

where $\phi$ is the prediction model randomly initialized for calculation and $e(\phi)$ is the mean prediction error on $\mathcal{D}_h$. We combine prediction loss and active reward with a regularization weight $\beta$:

$$r_t = r_t^i + \beta r_t^a. \tag{6}$$

During causal exploration, the agent keeps searching for causal informative data by maximizing the expected rewards, which is

$$\boldsymbol{a}_t^* = \arg\max_{\boldsymbol{a} \in \mathcal{A}} \mathbb{E}_{\tau \sim \pi}\left[\sum_t \gamma^t r_t\right], \tag{7}$$

where $\tau$ represents the trajectory generated by the exploration policy $\pi$ and $\gamma$ is the discount factor. Meanwhile, the world model optimizes to minimize the prediction loss. Since both of them contain the prediction error in equation (2) and (4), we can draw a conclusion that the learning of world models and causal exploration facilitate each other . Figure 3(d) shows an overview of causal exploration, details are shown in Algorithm 1.

### 3.1 CONVERGENCE ANALYSIS

In this subsection, we present a theoretical analysis of causal exploration in learning global world models. Learning to predict well in RL is typically a regression task. Thus, we consider the convergence rate of gradient descent in linear regression, formulated as

$$\min_w \quad e(w) = \|w^\top x - y\|_2^2, \tag{8}$$

where $x = [s, a] \in \mathbb{R}^{n+c}$ is the state-action sequence, $w \in \mathbb{R}^{n+c}$ is the linear weight of the world model and $y = {w^\star}^\top x$ is the ground truth value. Moreover, we make the following assumption:

**Assumption 3.1.** *$e$ is strong convex and smooth such that $\exists\ m > 0,\ M > 0$, for any $w \in \mathrm{dom}\ e$, we have $MI \succeq \nabla^2 e(w) \succeq mI$.*

Experiments are conducted to show that causal exploration is still helpful to actively and efficiently learn complex world models because the success of deep neural network remains unclear theoretically. Here we state the theorem under the smooth and convex assumption as follows.

**Theorem 3.1.** *Suppose Assumption 3.1 holds, and the density of causal matrix $D$ is $\delta$. If we initialize world model with $w_0$, update $w_t^c$ and $w_t$ with/without causal structure, denote $e^\star$ the optimal of $e(w)$, then we have*

$$e(w_t^c) - e^\star \leq \delta^t \left(e(w_t) - e^\star\right) \leq \frac{M}{2}\left[\delta\left(1 - \frac{m}{M}\right)\right]^t \|w_0 - w^\star\|_2^2. \tag{9}$$

**Remark**  This theorem establishes that the advantages of causal exploration is relevant to the sparseness of causal structure. The sparser the causal structure, the faster our method learns. When the causal matrix $D$ is a complete matrix ($\delta = 1$), causal exploration degenerates into non-causal prediction-based exploration. We prove Theorem 3.1 in appendix B.

---

**Algorithm 1** Task-agnostic Causal Exploration

---

1: **Initialize**:  Forward world Model $f$ with parameter $\boldsymbol{w}^c$
         Maximum training epoch $K$, trajectory length $T$, causal discovery period $N$
         Dataset $B_t$, exploration policy $\pi$ with memory buffer $\mathcal{M}$
2: **Output**: Forward world model $f$
3: **for** Episode $= 1, 2, \ldots, K$ **do**
4:     Initialize a random forward model $\phi$ and collect test set $\mathcal{D}_h$
5:     Set current time $t = 0$
6:     **while** $t < T$ **do**
7:         Choose action $\boldsymbol{a}_t = \pi(\boldsymbol{s}_t)$ and predict next state $\hat{\boldsymbol{s}}_{t+1} = f(\boldsymbol{s}_t, \boldsymbol{a}_t)$
8:         Obtain $\boldsymbol{s}_{t+1}$ from environment and calculate prediction reward $r_t^i$
9:         Update model $\phi$ and calculate active reward $r_t^a$ on $\mathcal{D}_h$
10:        Store $(\boldsymbol{s}_t, \boldsymbol{a}_t, r_t, \boldsymbol{s}_{t+1})$ into $\mathcal{M}$, store $(\boldsymbol{s}_t, \boldsymbol{s}_{t+1})$ into $B_t$
11:        Do online causal discovery on top-$k$ of $B_t$ per $N$ steps and get causal graph $\mathcal{G}$
12:        Train world model under $\mathcal{G}$ on $B_t$ and update $\boldsymbol{w}_i^c$
13:        Train exploration policy on $\mathcal{M}$ and update $\pi$
14:     **end while**
15: **end for**
16: **return**  Latest forward world model $f$

---

## 4 EXPERIMENTS

To verify the effectiveness of causal exploration in complex scenarios, we conduct several experiments both on synthetic dataset and real world applications, namely the traffic light control task and MuJoCo control suites (Todorov et al., 2012). In all of these settings, we share network parameters except for the last layer which is separately decomposed. Note that the form of intrinsic reward is not our primary focus, we use regression loss to train the agent and world model for simplicity. Besides, causal exploration with some other forms of prediction loss are included in appendix E.

We compare our proposed causal exploration with prediction-based exploration methods without leveraging causal information, including Curiosity-Driven Learning (Burda et al., 2018a) and Plan2Explore (Pathak et al., 2019). The intrinsic reward in Curiosity-Driven Learning is equation (4). Plan2Explore trains an ensemble of prediction models and uses the variance over forward predictions of the these ensemble members as its intrinsic reward. We also conduct comparative experiments with methods that attempt to gather causal information during exploration. CID (Seitzer et al., 2021) only detects causal influence on local action nodes, we extend their work to causal discovery between state transitions. Causal discovery and model learning are designed as two modules that facilitate each other in causal exploration. Unlike our work and CID, ASR (Huang et al., 2022) embeds structural constraints into the world model by employing a causal mask and train them together. Evaluation methods of these learned models include prediction errors during exploration and zero-shot performance on downstream RL tasks.

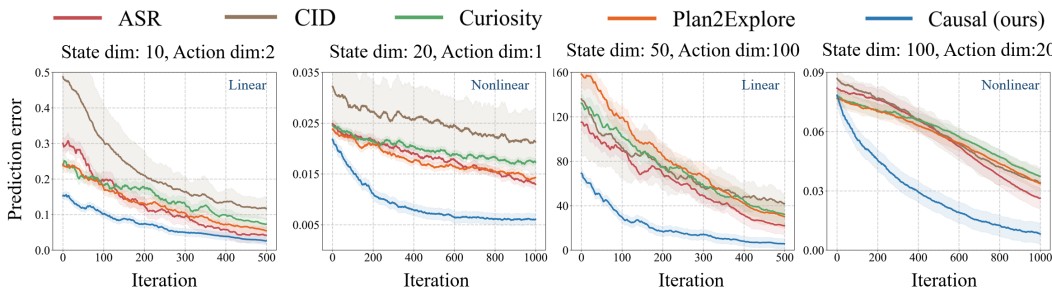

Figure 4: Prediction errors on synthetic environment with large state and action spaces.

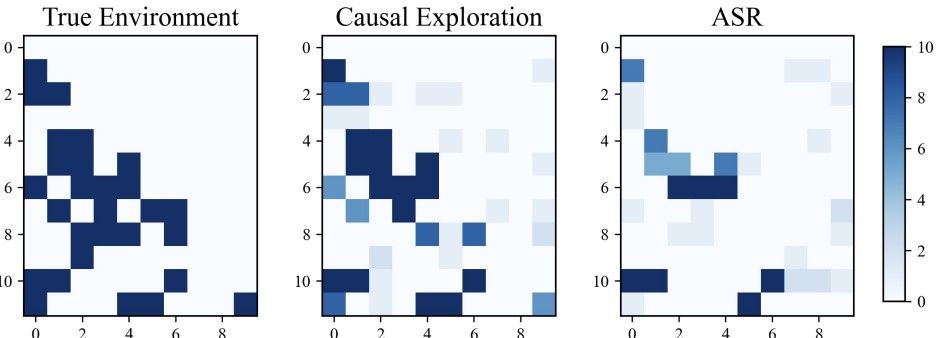

Figure 5: A comparison of the discovered causal matrix with different methods.

## 4.1 SYNTHETIC DATASET

**Environments**  We build our simulated environment following Deep Kalman Filters (Krishnan et al., 2015), a generative model of state and action sequences. Whenever the agent takes an action $a_t$ based on current state, the environment provides feedback $s_{t+1}$ at the next time. We denote the generative environment as

$$s_1 \sim \mathcal{N}(\mathbf{0}, I)$$
$$s_t \sim \mathcal{N}(h(s_{t-1}, a_{t-1}), \Sigma), \tag{10}$$

where $\Sigma$ is the covariance matrix and $h$ is the mean value as the ground truth transition function implemented by deep neural networks under causal graph $\mathcal{G}$. Specifically, the linear condition consists a single-layer network and the nonlinear function is three-layer MLPs with sigmoid activation.

Given that the sparsity of the causal graph is an important factor affecting the performance of our method, we choose to demonstrate the superiority of our approach on relatively low density causal structures. That is, $\mathcal{G}$ is generated by randomly connecting edges with a probability of $p$. Such sparse causal structures allow us to evaluate the ability of our method to accurately learn world models with limited data, which is a common scenario in many real-world applications.

**Results**  We first validate the success of our online causal discovery method in improving the efficiency. As can be seen in Figure 2(a), the execution time of the PC algorithm exhibits a noticeable inflection when the sample size is around 350. This provides a valuable reference for determining an appropriate value for the selection number $k$. Figure 2(b) further verifies this idea because the curve of times of KCI-test performed during PC illustrates a similar change. Additionally, it reveals that the increase in the execution time for PC algorithm is mainly due to growth of KCI-test times, especially when the depth of the condition set is 1. Figure 2(c) illustrates the corresponding ROC curves of different selection methods including Uniform Sampling, K-center Features (Nguyen et al., 2017), Hardest Sampling (Aljundi et al., 2019) and the Coreset Selection method (Yoon et al., 2021) we use. We observe that our online method leads to a remarkable quality improve of the causal discovery compared with no sampling while no other baseline could do this. This can be attributed to the successful removal of redundant and noisy samples.

After that, we conduct 10 experiments on the synthetic dataset and take the average value to reduce the impact of randomness. Figure 4 are prediction errors of the world models during exploration. Causal exploration achieves lower prediction errors with fewer data, which outperforms baselines in all of these environments. Note that the larger the dimension and the sparser the causal structure, the

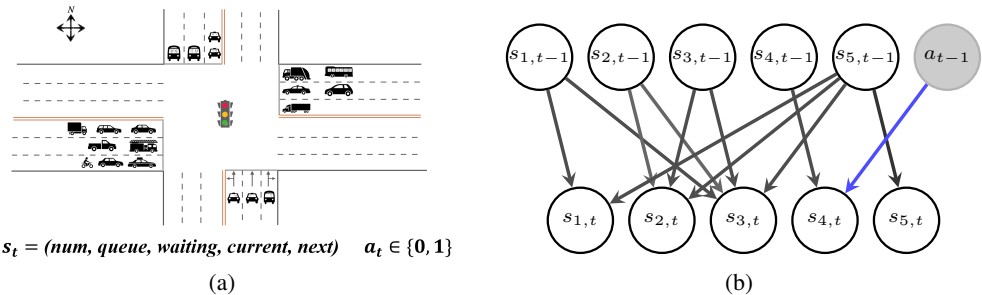

Figure 6: (a) Illustration of the traffic-signal-control environment. (b) Causal graph learned through exploration.

more prominent the advantages of our proposed method over non-causal method will be. Moreover, Figure 5 illustrates a concrete example of the causal matrix discovered by different methods, where the depth of the color indicates the number of times each edge is detected as a causal connection. The results reflect that the causal matrix learned through causal exploration obtains a smaller bias from the ground-truth matrix than that of ASR, an explanation for better performances.

We also observe that many of the prediction error curves have a different $y$ intercept, suggesting that the causal methods are better even at iteration $0$. This is because causal exploration exclusively incorporates the parent nodes and eliminate extraneous input information when predicting future states, a departure from prior methodologies where these nodes were treated and initialized on par with parent nodes. Consequently, even under random weight initialization for the world model, the prediction error of causal method are better at iteration $0$. This initial advantage is also tied to the sparsity of the causal graph as in Theorem 3.1. More implementation details and results are given in appendix C.1. Moreover, we consider challenging scenarios where the agent is start with wrong causal beliefs and encounters sudden structural changes in appendix C.2 and C.3.

## 4.2 REAL WORLD APPLICATIONS

**Application to Traffic Signal Control Task** Traffic signal control is an important means of mitigating congestion in traffic management. Compared to using fixed-duration traffic signals, a RL agent learns a policy to determine real-time traffic signal states based on current road conditions. The state observed by the agent at each time consists of five dimensions of information, namely the number of vehicles, queue length, average waiting time in each lane plus current and next traffic signal states. Action here is to decide whether to change the traffic signal state or not. For example, consider traffic signal is red at time $t$, if the agent takes action $1$, then it will change to green at the next time $t + 1$, otherwise it will remain red. The traffic environment used in our experiment is a three-lane intersection in IntelliLight (Wei et al., 2018), illustrated in 6(a).

Table 1: Performance of downstream policy learning in traffic signal control task.

| Model | Reward | Duration | Queue Length | Vehicles |
|---|---|---|---|---|
| ASR | -1.698 | 8.23 | 0.915 | 312 |
| CID | -4.356 | 12.45 | 3.619 | 288 |
| Curiosity | -3.505 | 10.92 | 1.550 | 304 |
| Causal World Model(ours) | **-1.366** | **7.21** | **0.301** | **314** |
| *True Environment* | *-1.007* | *7.61* | *0.112* | *316* |

During task-agnostic causal exploration, the agent first explores to efficiently learn a world model under intrinsic motivation. After that, the agent is provided with downstream tasks and needs to solve the tasks based on prediction of the learned world model instead of environment interactions. Table 1 lists the performance of different learned world models on downstream policy learning. The reward here is a combination of several terms including sum of queue length and so on, defined in equation (3) in IntelliLight. Duration refers to average travel time vehicles spent on approaching lanes (in seconds). Queue Length is the sum of the length of waiting vehicles over all approaching lanes and Vehicles is the total number of vehicles that passed the intersection. We also train an agent in the true environment to verify the reliable of the learned models through task-agnostic exploration.

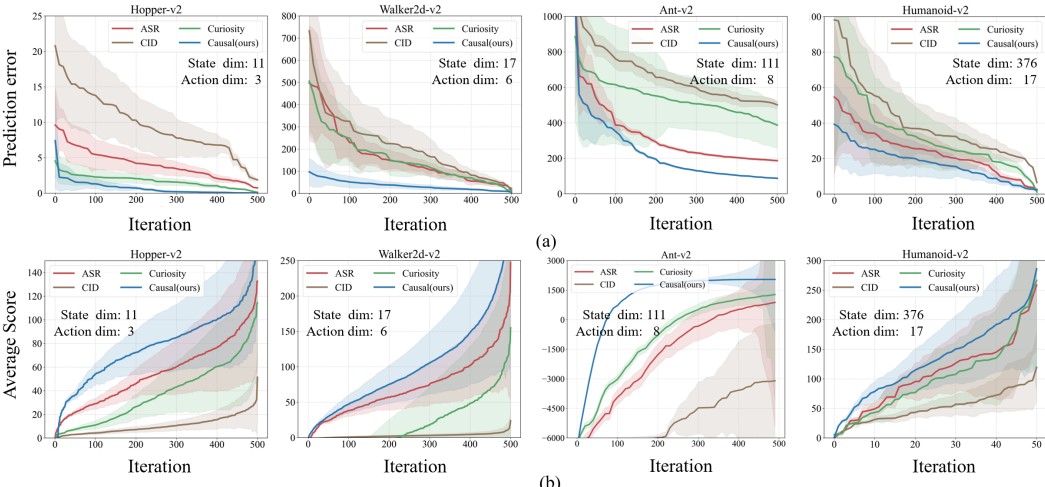

Figure 7: (a) Prediction errors in world model learning and (b) downstream policy learning performance on MuJoCo tasks.

The performance of the policy learned on the world models trained during causal exploration is comparable to, or even in some metrics better than, that of the true environment. The causal graph on state variables in traffic signal control task is shown in 6(b), comparison of the prediction errors during exploration and detailed analyses are included in appendix D.

**Application to MuJoCo Tasks** In addition to traffic light control scenario, we also evaluate causal exploration on challenging MuJoCo tasks. MuJoCo is a popular physics simulation engine used for modeling the motion and physical interactions of multi-joint robots in complex environments. In these tasks, state variables typically include robot joint positions, velocity information, and some other relevant factors, while actions are corresponding joint movements. The state-action dimensions of Mujoco tasks range from tens (Hopper-v2) to hundreds (Humanoid-v2). We use PPO algorithm (Schulman et al., 2017) for optimization during both the task-agnostic exploration and policy learning stages. Appendix E gives a detailed description of the hyperparameters for the Mujoco task.

As is depicted in Figure 7(a), the world model learned during causal exploration always performs lower prediction errors than others. This ability of our causal method stems from its focus on causal relationships and structural constraints. By selectively incorporating relevant parent nodes during state prediction, our approach maintains informed exploration even when exploratory signals decline, which prevents the agent from getting stuck in sub-optimal behaviors.

Such a good performance is also achieved in a shorter time. Task performances in Figure 7(b) also reflects the reliability of these world models. We see that our proposed method still works well when the state space is large. As long as there is a strong causal relationship between the observed state variables, and the causal discovery algorithm can accurately identify the corresponding causal structure, our causal exploration method is believed to be well applied to high-dimensional situations.

## 5 CONCLUSION

In this paper, we introduce causal exploration, a methodology designed to incorporate causal information from data for the purpose of learning world models efficiently. In particular, we employ causal exploration within the domain of task-agnostic reinforcement learning and design a sharing-decomposition schema to leverage causal structural knowledge for the world model. A series of experiments in both simulated environments and real world tasks including traffic light control and MuJoCo demonstrate the superiority of causal exploration, which highlight the importance of rich causal prior knowledge for efficient data collection and model learning. We would like to point out that this study primarily focuses on scenarios where we fully observe latent states rather than pixel inputs. Future research directions include considering complex scenarios such as unobserved latent state variables and designing better fault-tolerant mechanisms to enhance robustness. We believe that this work provides a promising direction for future efficient exploration.

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
