In the following, we first introduce the related works. After that, the proof of Theorem 3.1 is presented. Finally, more experimental details about the synthetic and real world applications are given.

## A  A DETAILED DESCRIPTION OF RELATED WORK

### A.1  EXPLORATION STRATEGIES IN REINFORCEMENT LEARNING

Reinforcement learning (RL) has emerged as a powerful tool for training intelligent agents to perform complex tasks, ranging from playing games like Go to autonomous driving and traffic signal control tasks (Silver et al., 2016; Tassa et al., 2018; Wei et al., 2018; Kiran et al., 2021). The success of RL largely depends on the ability of the agent to explore its environment, gather information and learn from experience. Various exploration strategies have been proposed to tackle the exploration-exploitation dilemma, especially in deep reinforcement learning (DRL). Traditional exploration strategies include $\epsilon$-greedy (Sutton & Barto (2018)), upper confidence bounds (Auer (2002)), Boltzmann exploration (Cesa-Bianchi et al. (2017)) and Thompson sampling (Russo et al. (2018)). However, the data collection and policy learning process are usually expensive due to hours of environment interactions, especially in large state spaces and sparse or even deceptive reward environments where discovering the optimal policy can be highly challenging.

Over recent decades, task-agnostic exploration strategies has been an active research area to tackle such challenge (Aubret et al., 2019). Existing methods focus on designing appropriate forms of intrinsic rewards, which can be broadly categorized into three types. (1) the number of times a state has been visited, which helps to guide the agent towards unexplored regions (Brafman & Tennenholtz, 2002; Bellemare et al., 2016; Machado et al., 2020). These count-based methods (Ostrovski et al., 2017; Zhao & Tresp, 2019) define an empirical count function to record the agent's behavior and guide it to rarely visited states. In adaptation of high-dimensional states, Tang et al. (2017) propose to use domain-dependent hash functions to map states before counting. (2) curiosity about the environment dynamics, which is usually formalized as prediction errors on the next state (Pathak et al., 2017; Burda et al., 2018b; Kim et al., 2018). Prediction-based exploration relies on the quality of state representation, Burda et al. (2018b) show that exploring in raw image space is inefficient and even random features given by a fixed neural network demonstrate competitiveness. (3) information gain, which aims to improve the agent's knowledge about the environment by maximizing the mutual information (Duan et al., 2016; Shyam et al., 2019). Recently, information gain is used by Achiam & Sastry (2017) and Sekar et al. (2020) as an excellent solution to the white-noise problem. Despite significant advancements in research on task-agnostic exploration strategies, prior methods pay little attention to the causal structure of the data.

### A.2  CAUSAL DISCOVERY

Over the years, several standard causal discovery benchmarks have been developed, including Bayesian methods (Heckerman, 2013), the Wermuth-Lauritzen algorithm (Lauritzen & Wermuth, 1989), PC (Spirtes et al., 2000) and IG (Verma & Pearl, 1990). These causal methods have been successfully applied in a wide range of domains, including biology (Ross, 2021), economics (Emirmahmutoglu & Kose, 2011), and social sciences (Blalock, 2017).

Recently, intervention-based causal reasoning (Jaber et al. (2020)) has attracted attention in reinforcement learning, which involves actively manipulating variables in a system to observe causal effects. However, it is infeasible to perform interventions in many real-world scenarios. As a result, observational causal discovery methods, such as constraint-based (Yu et al. (2016)) and score-based methods (Zhu et al. (2019)), are used to infer causal relationships from the data in these tasks.

Furthermore, extracting causal knowledge to help intelligent agents make interpretable decisions has also become a popular trend in task-specific reinforcement learning methods (De Haan et al., 2019; Ke et al., 2021). Molina et al. (2020) makes use of causal structures to restrict the action space during exploration. Jaber et al. (2020) involves actively manipulating variables in a system to observe the causal effects during exploration and Peng et al. (2022) proposes to automatically learn a hierarchical causal structure for subgoal-based policy training.

## A.3 ACTIVE LEARNING

Active learning is a method describing how to utilize data resources to improve the training efficiency. Typically, a subset of samples that the model mostly unfamiliar with are selected for training (Settles, 2009). Existing methods for active learning can be categorized into three classes. (1) In uncertainty approaches, probabilistic learning models such as entropy (Holub et al., 2008; Siddiqui et al., 2020) or margins (Balcan et al., 2007) are used to define uncertainty of the data. (2) Diversity algorithms choose data that best represent the whole distribution, which formulate the selection into optimization problems (Elhamifar et al., 2013; Wang et al., 2017) or apply clustering methods (Nguyen & Smeulders, 2004). (3) The key idea of expected error reduction methods (Roy & McCallum, 2001) is to select data points that would contribute the largest loss reduction to the model.

## B   PROOF OF THEOREM 3.1

Recall that we consider solving a linear regression problem under $m$ strong convex and $M$-smooth. MSE loss function $e(w)$ in the formulation $\|w^\top x - y\|_2^2$ and $y = {w^*}^\top x$ is minimized by gradient descent method with

$$w_{t+1} = w_t - \alpha \nabla e(w_t) \tag{11}$$

where $\alpha$ is the corresponding stepsize.

Denote $w_t^c$ as the network parameters taking causal constraints with respect to $w_t$ that dose not gather causal information. Theorem 3.1 shows that causal exploration gets a prediction error bound $\delta^t$ times lower at time $t$, where $\delta$ is a density measurement of causal matrix $D$.

Below, we present a two-step proof for the convergence of causal exploration. Lemma B.1 provides an upper bound for convergence without using any causal information. Lemma B.2 demonstrates that utilizing causal structure information results in $w_t^c$ being closer to the optimal value $w^*$ compared to $w_t$ at same update steps. Combining Lemma B.1 and Lemma B.2, we derive the convergence rate for causal exploration.

**Lemma B.1.** *Suppose $e(w)$ is $m$-strongly convex and $M$-smooth. We have*

$$\|w_t - w^*\| \le (1 - \frac{m}{M})^t \|w_0 - w^*\|. \tag{12}$$

*Proof.* According to gradient descent method (11), we get

$$\|w_t - w^*\|^2 = \|w_{t-1} - \alpha \nabla e(w_{t-1}) - w^*\|^2$$
$$= \|w_{t-1} - w^*\|^2 - 2\alpha \nabla e(w_{t-1})(w_{t-1} - w^*) + \alpha^2 \|\nabla e(w_{t-1})\|^2.$$

By strong convexity

$$\nabla e(w)(w - w^*) \ge e(w) - e(w^*) + \frac{m}{2}\|w - w^*\|^2, \tag{13}$$

we further obtain

$$\begin{aligned}
\|w_t - w^*\|^2 &\le \|w_{t-1} - w^*\|^2 - 2\alpha(e(w_{t-1}) - e(w^*) + \frac{m}{2}\|w_{t-1} - w^*\|^2) \\
&\quad + \alpha^2 \|\nabla e(w_{t-1})\|^2 \\
&= \|w_{t-1} - w^*\|^2 - 2\alpha(e(w_{t-1}) - e(w^*)) - \alpha m\|w_{t-1} - w^*\|^2 \\
&\quad + \alpha^2 \|\nabla e(w_{t-1})\|^2 \\
&\le \|w_{t-1} - w^*\|^2 - 2\alpha(e(w_{t-1}) - e(w^*)) - \alpha m\|w_{t-1} - w^*\|^2 \\
&\quad + 2\alpha^2 M(e(w_{t-1}) - e(w^*)) \\
&\le \|w_{t-1} - w^*\|^2 - \alpha m\|w_{t-1} - w^*\|^2 + 2\alpha(\alpha M - 1)(e(w_{t-1}) - e(w^*)).
\end{aligned} \tag{14}$$

Consider $\alpha = \frac{1}{M}$, we get

$$\|w_t - w^*\|^2 \le (1 - \frac{m}{M})\|w_{t-1} - w^*\|^2. \tag{15}$$

Using the above equation repeatedly, we obtain

$$
\begin{aligned}
\|w_t - w^*\|^2 &\leq (1 - \frac{m}{M})\|w_{t-1} - w^*\|^2 \leq (1 - \frac{m}{M})^2\|w_{t-2} - w^*\|^2 \\
&\leq \cdots \leq (1 - \frac{m}{M})^t\|w_0 - w^*\|^2.
\end{aligned}
\tag{16}
$$

$\square$

**Lemma B.2.** *Suppose $w_t^c \in \mathbb{R}^{n+c}$ and $w_t \in \mathbb{R}^{n+c}$ are the network parameters with/without causal structure respectively and $w^*$ is the optimum. It holds that*

$$
\|w_t^c - w^*\|^2 \leq \delta_t \|w_t - w^*\|^2.
\tag{17}
$$

*Proof.* According to the sharing-decomposition schema, we have

$$
w_t^c = D \odot w_t,
\tag{18}
$$

where $D \in \{0, 1\}^{(n+c)}$ is the causal matrix. Hence, we rewrite $w_t$ as

$$
w_t = w_t^c + w_t'.
\tag{19}
$$

Note that we have $w_t^c \perp w_t'$ and $w^* = D \odot w^*$. Then we obtain

$$
\begin{aligned}
\|w_t - w^*\|^2 &= \|w_t^c - w^* + w_t'\|^2 \\
&= \|w_t^c - w^*\|^2 + \|w_t'\|^2 \\
&\geq (1 + \rho_t)\|w_t^c - w^*\|^2,
\end{aligned}
\tag{20}
$$

where $\rho_t \in [0, +\infty)$ is the lower bound of the ratio between $\|w_t'\|^2$ and $\|w_t^c - w^*\|^2$ whose value is related to the sparsity of causal matrix $D$. By setting $\delta_t = \frac{1}{1+\rho_t}$, we complete the proof of Lemma B.2. $\square$

The convexity of $e(w)$ implies

$$
e(w_t^c) - e(w^*) \leq \frac{M}{2}\|w_t^c - w^*\|^2.
\tag{21}
$$

By applying Lemma B.1 and Lemma B.2 into (21) and denote

$$
\delta = \max\{\delta_0, \delta_1, \ldots, \delta_t\},
\tag{22}
$$

we can obtain Theorem 3.1.

## C  SYNTHETIC ENVIRONMENT

### C.1  MORE EXPERIMENT DETAILS

**Implementation**  Double DQN (Van Hasselt et al., 2016) is used to train the exploration policy of agents, where both the evaluation network and the target network are three-layer fully connected networks with relu activations. The corresponding world models are designed as MLPs without/with activation function in linear/nonlinear module respectively, with 32 and 8 hidden nodes. We use Adam optimizer and MSE loss to learn above models.

Table 2: Results of different selection methods on online causal discovery.

| Method | No Sampling | Uniform | K-center | Hardest | Coreset |
|---|---|---|---|---|---|
| AUC | $0.907 \pm 0.084$ | $0.892 \pm 0.065$ | $0.891 \pm 0.175$ | $0.816 \pm 0.353$ | $\mathbf{0.969 \pm 0.016}$ |
| Precision | $0.953 \pm 0.074$ | $0.914 \pm 0.079$ | $0.942 \pm 0.089$ | $0.892 \pm 0.081$ | $\mathbf{0.970 \pm 0.064}$ |
| F1-score | $0.831 \pm 0.094$ | $0.842 \pm 0.104$ | $0.890 \pm 0.139$ | $0.852 \pm 0.067$ | $\mathbf{0.928 \pm 0.096}$ |
| Time | $4.360 \pm 2.888$ | $0.880 \pm 0.453$ | $0.909 \pm 0.414$ | $1.389 \pm 0.760$ | $\mathbf{0.379 \pm 0.267}$ |

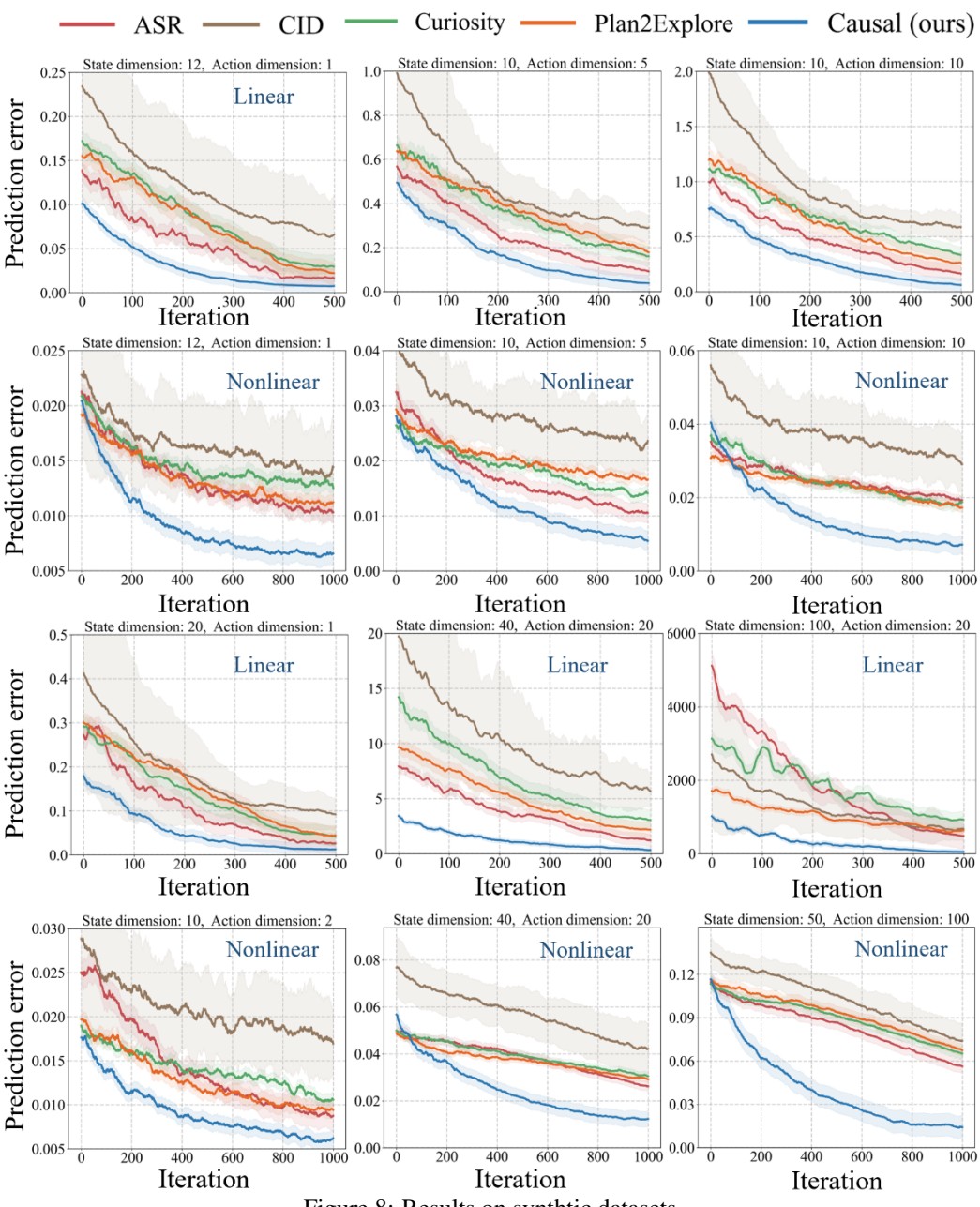

Figure 8: Results on synthtic datasets.

**Efficient Causal Discovery**   To speed up PC algorithm with KCI-test, we design an efficient online causal discovery using selection methods. Tables 2 summarizes the performance of different sampling methods in our synthetic environments. These exciting experimental results demonstrate the superiority of the Coreset Selection method we used in improving the efficiency of causal discovery. The selection size $k$ is empirically set as $350$ and kept the same among different methods except for the no sampling method that takes all of the data points.

**Causal Exploration Experiments**   The causal matrix in our synthetic environment is set as a lower triangular matrix whose elements are generated from uniform distribution $[-8, 8]$. After that, for each edge in the graph, we randomly drop it out with a empirically chosen probability $1 - p = 0.8$. Besides, the covariance $\Sigma$ is a diagonal matrix whose elements are randomly generated from $[0, 0.1]$. For both linear and nonlinear conditions, we have conducted sufficient experiments. Figure 8 shows the remaining experimental results that are not fully presented in the main body.

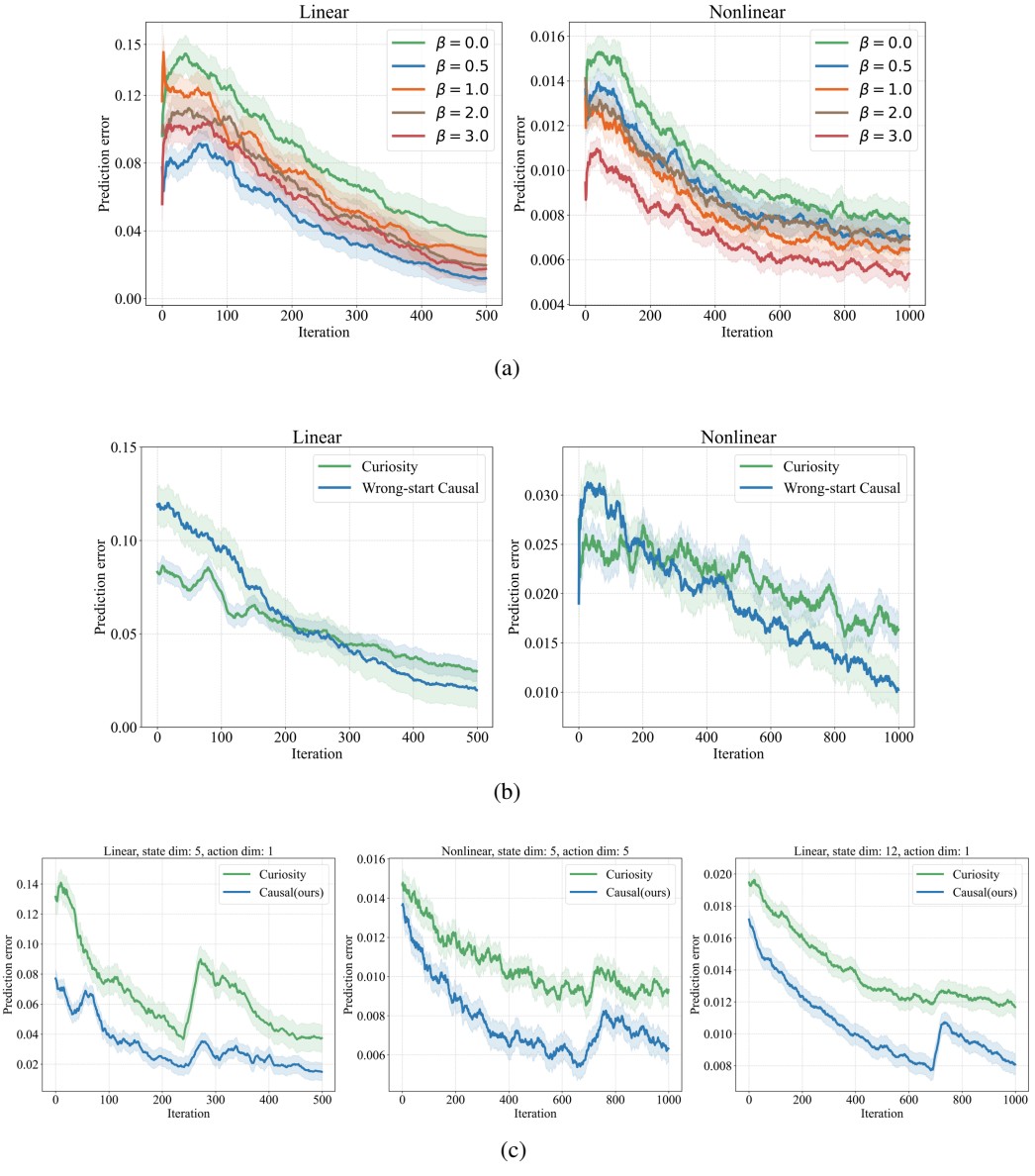

Figure 9: (a)Prediction errors of causal exploration for different $\beta$; (b) Performance on underestimation scenarios; (c) Scenarios with structural changes.

In order to enhance the agent's sensitivity to causal informative data, we design a novel form of active reward. Here we investigate the impact of $\beta$ in this new intrinsic reward formulation on causal exploration which is formulated as $r_t = r_t^i + \beta r_t^a$ in equation (6). Figure 9(a) illustrates the evolution of prediction error over training time for different values of $\beta$. We see that incorporating active reward for exploration continuously improves the performance of the world model. We set $\beta$ as $0.5$ and $3$ for linear and nonlinear environments according to the results in Figure 9(a), respectively.

## C.2 GENERALIZATION TO UNDERESTIMATION SCENARIOS

In some data-hungry scenarios, there may be insufficient data for causal discovery, leading to underestimation of the causal structure, which makes continuous data collection and causal structure correction important components. In other words, the causal structure inferred from causal discovery algorithms may deviate from the ground truth ones, which is particularly prone to occur under conditions of limited sample size or during the initial stages of exploration. Consequently, we conducts evaluation of our proposed algorithm's performance when the estimated causal structure exhibits in-

sufficiencies or redundancies, a scenario termed "underestimation". To highlight the advantages of our sharing-decomposition schema in addressing such problems, we deliberately provide the agent with completely wrong causal information. An erroneous causal graph $\mathcal{G}'$ is supplied to the agent during the initial time steps, namely $t < N$ where $N$ is the period for causal discovery. We introduce perturbations to the true graph $\mathcal{G}$ by randomly eliminating or adding edges with a probability of $p' = 0.8$, generating $\mathcal{G}'$. Figure 9(b) shows corresponding performance of causal exploration on synthetic data. Indeed, the agent exhibits an impressive ability to correct its causal exploration trajectory in the face of misdirection. Nonetheless, it is worth noting that such a schema only serves as a technique to mitigate the impact of underestimation. The reliability of causal discovery algorithms is the essential guarantee for causal exploration.

### C.3 GENERALIZATION TO SCENARIOS WITH CAUSAL STRUCTURAL CHANGES

In real-world scenarios, causal structure between variables can often change due to sudden disturbance. For instance, causal relationships between economic variables like stock prices, interest rates, and inflation can be subject to rapid changes caused by market crashes or policy changes.

To evaluate the effectiveness of our approach in handling such mutation, we conduct experiments in a scenario where the causal structure change randomly once. We use our simulation model to generate the data and compare our method to a non-causal approach. Figure 9(c) illustrates the advantages of our approach in tackling such a challenging task.

Our sharing-decomposition schema enables the agents to quickly adapt to structural changes and make appropriate adjustments. This also demonstrates the robustness of our method, which allows for timely correction of errors in the causal structure. By sharing the same decomposition modules across different time steps and tasks, our method can effectively leverage previous knowledge and transfer it to new situations, while also being flexible enough to accommodate changes in the causal structure. In addition, the ability to adapt to changing causal structures can improve the generalization ability of our method, making it more applicable to a wider range of real-world tasks.

In our future research, we plan to expand our work to situations where changes occur within the model. In these cases, during the model learning phase, it becomes crucial to effectively detect these changes and promptly update the model. Additionally, when it comes to policy learning, a key challenge is determining the most suitable model to utilize. We may encounter entirely new models that have not been encountered before, adding an additional layer of complexity to our research.

## D  TRAFFIC SIGNAL CONTROL

Following the work in IntelliLight (Wei et al., 2018), the traffic environment in our experiment is a three-lane intersection. Table 3 gives a detail description.

Table 3: Traffic Dataset Description

| Traffic flow setting | Directions | Arrival Rate (cars/s) | | Duration ($\times 10^3$ s) |
|---|---|---|---|---|
| | | Mean | Std | |
| Complex traffic | East-West | 0.211 | 0.023 | 216 |
| | South-North | 0.155 | 0.030 | |

**Experiment details and analysis**   We first purely use prediction-based causal exploration to learn forward dynamic world models under the same traffic environment in IntelliLight. Then, the agent learn a policy for traffic signal control task in our learned world models, which avoids the high-cost interaction with real traffic environment. For consistency and easy comparison, we use the same DQN network from IntelliLight to train our causal exploration agent. After that, we solve traffic signal control task in our world model with no environment interaction in a zero-shot manner. Figure 10 visualizes the prediction errors of different models during the exploration process.

The corresponding causal graph is shown in Figure 6(b). As is illustrated, the state of the traffic signal at the next time step $s_{4,t}$ is causally linked to the previous state $s_{4,t-1}$ and action $a_{t-1}$, which is in line with the definition of traffic signal control tasks. The queue length $s_{2,t}$ is determined by

previous queue length $s_{2,t-1}$ and waiting time $s_{3,t-1}$ plus the traffic state $s_{5,t-1}$. Factors influencing the waiting time $s_{3,t}$ include the number of vehicles $s_{1,t-1}$ and the queue length $s_{2,t-1}$. These results align well with the common-sense reasoning.

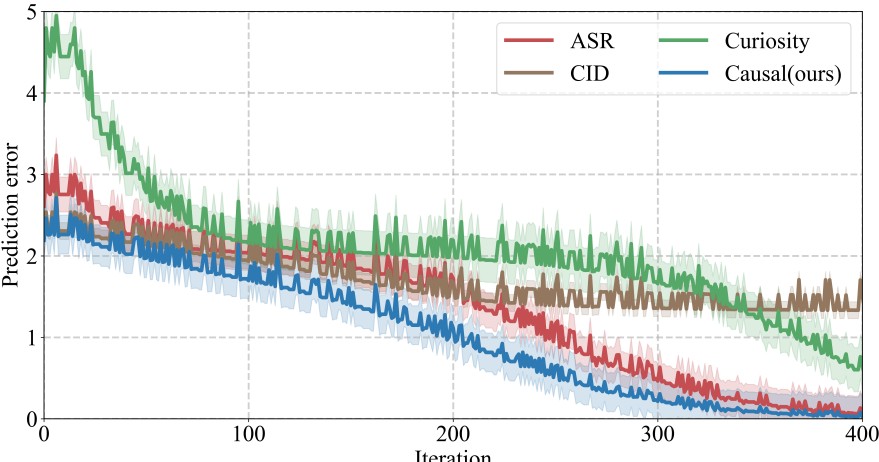

Figure 10: Prediction errors in traffic light control.

## E   MORE RESULTS OF MUJOCO TASKS

We adopt the hyperparameters from Table 3 of PPO (Schulman et al., 2017) with a trajectory length of $2048$, an Adam stepsize of 3e-4, a minibatch size of 64, a discount factor ($\gamma = 0.99$), a GAE parameter ($\lambda = 0.95$), and a clipping parameter ($\epsilon = 0.2$). Both the actor-critic network and the world model are 2-(hidden)-layer neural networks, consisting of 256 and 64 hidden nodes respectively. Activation functions are Tanh and ReLU here. Performance of causal exploration on some other MuJoCo tasks are provided in Figure 11. Predictions given by world models under causal structural constraints are more accurate and stable than those of other methods. The learned world model of causal exploration provides the agent with more information in the following policy learning stage, resulting in higher scores achieved in a shorter time.

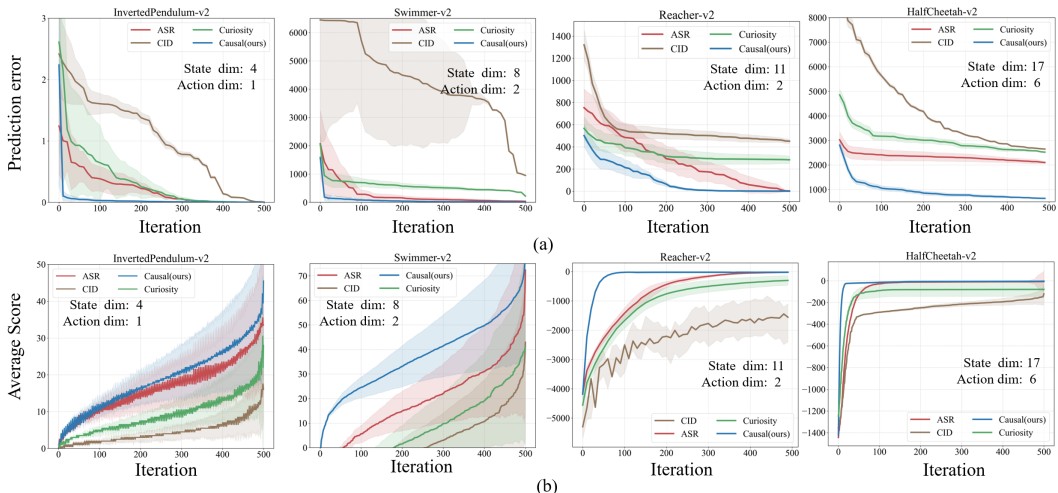

Figure 11: Application to some other MuJoCo tasks.

We also conduct several experiments to test the performance of causal exploration with a different form of intrinsic reward. To be specific, we formulate our world model as $\mu_{w^c}(\boldsymbol{s}_t, \boldsymbol{a}_t), \sigma^2_{w^c}(\boldsymbol{s}_t, \boldsymbol{a}_t)$ to model the transition probability as $p(\boldsymbol{s}_{t+1} \mid \boldsymbol{s}_t, \boldsymbol{a}_t) \sim \mathcal{N}(\boldsymbol{s}_{t+1}; \mu_{w^c}, \sigma^2_{w^c})$. Then, the negative log-likelihood is used both for the world model learning and causal exploration, which is a replacement

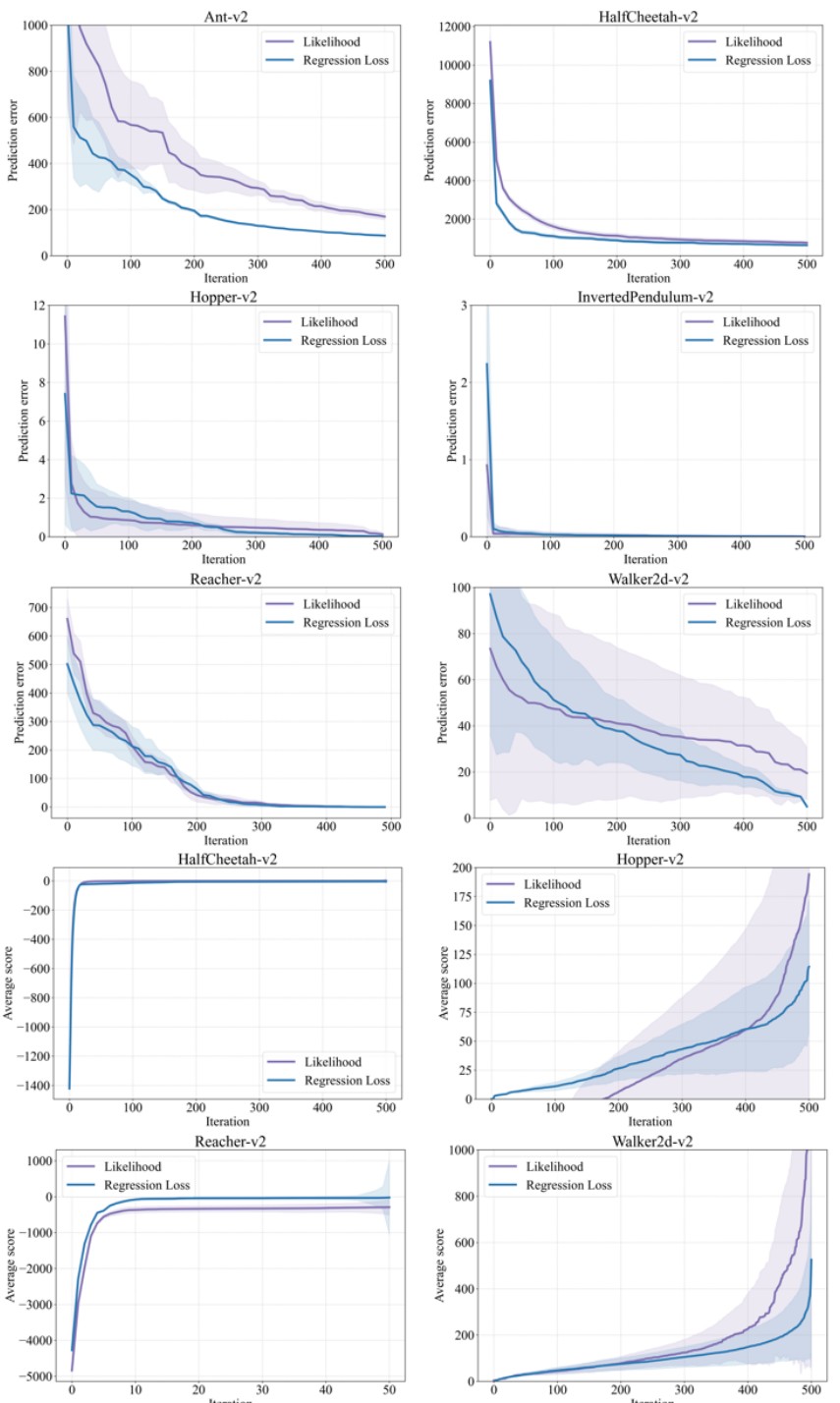

Figure 12: Performance of causal exploration with different forms of intrinsic rewards.

for equation (2) and (4), and is formulated as:

$$L_{(\mu_{w^c}, \sigma^2_{w^c})} = \frac{(\boldsymbol{s}_{t+1} - \mu_{w^c}(\boldsymbol{s}_t, \boldsymbol{a}_t))^2}{2\sigma^2_{w^c}(\boldsymbol{s}_t, \boldsymbol{a}_t)} + \frac{1}{2} \log \sigma^2_{w^c}(\boldsymbol{s}_t, \boldsymbol{a}_t), \quad r^i_t = \frac{\eta}{2} L_{(\mu_{w^c}, \sigma^2_{w^c})}. \tag{23}$$

Corresponding results are shown in Figure 12. However, various forms of intrinsic rewards don't exhibit significant differences in performance. In most tasks, the introduction of an additional co-variance network even lead to performance not as favorable as when directly using regression loss.