# OpenReview forum: "Boosting Efficiency in Task-Agnostic Exploration Through Causal Knowledge"
_ICLR.cc/2024/Conference — Submitted to ICLR 2024_

### Official Review · Reviewer_GFa4 · 2023-10-30

**Soundness:** 2 fair
**Presentation:** 2 fair
**Contribution:** 1 poor
**Rating:** 3
**Confidence:** 4

**Summary:**

The paper first introduces a framework that (a) discovers causal relationships between state and action variables, and simultaneously, (b) trains the world model that respects the discovered causal graph (this framework is called *causal dynamics learning* in the literature). For (a), the authors rely on conditional independence tests (CITs). They are widely used for causal discovery, but the computational cost grows exponentially with respect to the sample size. For this, the authors utilize a coreset sampling strategy for efficient CIT. For (b), the dynamics model adheres to Markov factorization guided by the discovered causal graph, i.e., $p(s'\mid s, a)=\prod_j p(s'_j\mid Pa(s_j'))$. This requires different models for each state variable, so the authors introduce weight-sharing networks to reduce model parameters.

Along with this causal dynamics learning framework, the paper proposes a causal exploration which is defined as the combination of the (i) sample prediction error and (ii) improvement of the average prediction error on the test set. The authors also provide convergence analysis. Experimental results demonstrate that the proposed causal exploration efficiently discovers causal relationships and outperforms previous exploration strategies.

**Strengths:**

- The paper deals with an important problem. The paper is also easy to follow and well-written.
- The idea of employing the coreset selection strategy of Yoon et al., (2022) for efficient conditional independence tests is interesting. The paper also provides a convergence analysis.
- Experiments are made extensively from synthetic environments to complex tasks, e.g., traffic signal control tasks.

**Weaknesses:**

First, the authors did not recognize the highly relevant works in the line of **causal dynamics learning in RL** [1-3]. These causal dynamics models aim to discover causal relationships and train dynamics models accordingly, which is exactly what Sec. 2 is about. For example, [3] also utilized CITs to determine whether the edges exist or not, while [1] used a score-based method. Efficient dynamics model architecture is also proposed and discussed in [2]. Moreover, [2] also proposed an exploration term to boost causal dynamics learning, but the manuscript does not discuss nor mention these works at all, which seems a critical issue. Unfortunately, in its current form, it is hard to assess the novel development of this paper compared to prior works. Second, I also have a concern regarding the presentation and experiments. The points that need further clarification or discussion are listed below.

**References**

[1] Wang et al., “Task-independent causal state abstraction”, NeurIPS Robot Learning Workshop 2021.

[2] Wang et al., “Causal dynamics learning for task-independent state abstraction”, ICML 2022.

[3] Ding et al., “Generalizing goal-conditioned reinforcement learning with variational causal reasoning”, NeurIPS 2022.

**Questions:**

- Given that the edges follow the temporal order and there are no instantaneous edges, it is basically applying a conditional independence test for every possible candidate edge. Calling it a PC algorithm is somewhat misleading.
- Score-based approach is more efficient compared to constraint-based methods for causal discovery. Justification and discussions would be appreciated.
- What is the depth in Fig 2 (b)?
- In Eq. 2-3, what is the difference between $w$ and $w^c$?
- How is the separate prediction model $\phi$ updated? What is $\phi_t$ in Eq. 5? Also, why do we need a separate model, instead of the forward dynamics model?
- In Mujoco, the causal relationship seems dense (i.e., fully-connected), but the proposed method outperforms other baselines. Given that the proposed causal exploration degenerates into non-causal exploration when the causal connection is fully connected (as described in Sec 3.1), the performance gain shown in Fig. 7 needs further explanation.
- What is the value of $\beta$ in Traffic Signal Control and Mujoco?

**Minor comments and typos**

- Plot headings in Fig. 2 (a) and (c): “PC discovery” is a bit awkward. It should be PC algorithm or causal discovery.
- In Eq. 1, the gradient operator for $f_w(b_t^p)$ should be a sample gradient, not the average gradient.
- There should be spaces between words and parentheses (like this).

---

> ### Author Response · Authors · 2023-11-15
> **Thanks for the review (part 1)**
>
> We greatly appreciate the reviewer's thorough and constructive comments. We attempt to address all the concerns in the following.
>
> **Weakness**$\quad$After carefully reading the mentioned works, it holds true that there exist several papers that are relevant to ours. We apologize for any oversight in our previous acknowledgment of related works and appreciate the opportunity to clarify this and will appropriately cite these papers and make comparisons. However, we also want to clarify that the exploration term used in [1,2] is actually the same as the ASR method, which has alreadby been taken as one of our baselines. In addition, the conditional independence test in [3] uses all of the coming data, while we selectively collect representative data points during exploration in an incremental way (equation 1).
>
> **Questions**
>
> **PC algorithm**$\quad$Thank you for point this out. We acknowledge that a more rigorous phrasing would be, "We extend PC to time series data with time-lagged causal relationships". We will promptly address this clarification in our latest version.
>
> **Score-based approach** $\quad$Thank you for your suggestion. We will incorporate the justification and discussions comparing our approach with score-based methods in the latest version. However, we cannot fully agree with the statement, "Score-based approach is more efficient compared to constraint-based methods for causal discovery." As mentioned in the Q3 section of the referenced work [3], under the MDP and Causal Faithfulness assumptions, "Based on the performance of the Score model summarized in Table 1, we also conclude that it (the score-based method) is not as good as our constraint-based method and has a large variance due to the unstable learning of the causal graph." This observation similarly applies to our work.
>
> **Depth in Fig 2 (b)**$\quad$"depth" here refers to the cardinality of the conditioning set required for conditional independence tests.
>
> **Difference between $w$ and $w^c$**$\quad$As stated in Section B of the supplementary materials, we " denote $w^c_{t}$ as the network parameters incorporating causal constraints with respect to $w_t$ that does not gather causal information". Sorry for any inconvenience caused in your reading and we will modify the corresponding expression in the main paper.
>
> **Active reward**$\quad$The method in [1] trains a strategy based on active reward to determine whether to annotate the incoming data for a supervised classifier. If the active reward value is positive, it indicates that the annotated data is helpful for the training process. Similarly, we use active rewards to encourage agents to explore toward data that is helpful for training. Therefore, active rewards are used to measure the training quality. To maintain consistency in the order of prediction loss, we need to ensure that the agent can receive prediction errors and active rewards at every time step $t$. Therefore, $t$ in equation 5 refers to the time step. However, the optimization of the world model $f$ doesn't occur at every time step $t$, so we need to separately extract a dynamics model $\phi_t$ for calculating the active reward. We discuss this in detail in Section C.1 of the supplementary materials. More information on the intuition of active reward can be found in [1].
>
> [1] Meng Fang, Yuan Li, and Trevor Cohn. Learning how to active learn: A deep reinforcement learning approach. arXiv preprint arXiv:1708.02383, 2017.

---

> ### Author Response · Authors · 2023-11-15
> **Thanks for the review (part 2)**
>
> **Mujoco** Thank you for the thoughts. A detailed explanation of the performance gain in Figure 7 is outlined as follows:
>
> * Complexity of Causal Relationships in Mujoco: As shown in prior works [1-4], the dynamics in many real world problems is typically sparse. Therefore, it is necessary to clarify whether this is also the case in Mujoco. However, an extensive review of existing literature did not yield documentation or theoretical works describing the authentic causal relationships in Mujoco tasks.
>
> *  Causal Structure Identified by PC Algorithm: In our work, the causal structure identified by the PC algorithm appears to be relatively not that dense, a trend supported by empirical results such as prediction error and performance gain. Unfortunately, due to the absence of benchmark works, we cannot guarantee the complete correctness of the identified causal graph. We appreciate your valuable feedback that points this out, and we will include a discussion on the correctness of identified causal relationships in Mujoco tasks in our latest version, as demonstrated in simulated data (Figure 5) and the traffic light control task (Figure 6).
>
> * Two-Stage Approach and Model Learning Impact: Our method consists of two stages: model learning and policy learning. Policy learning is based on the model learned in the first stage. Therefore, the accuracy of model learning significantly influences policy learning. In the first stage, we use a limited amount of data for model learning, which might be sufficient for causal models but inadequate for non-causal models, as reflected in Figure 7(a). This explains the results in Figure 7(b).
>
> **The value of $\beta$**$\quad$We set $\beta$ as $3$ according to the nonlinear results in Section C.1 of the supplementary materials.
>
> **Minor comments and typos**
>
> Thank you very much for pointing out these typos. We will carefully review and address them in the latest version.

---

> > ### Comment · Reviewer_GFa4 · 2023-11-22
> >
> > Thank you for your response. I have read the comments from the other reviewers, and I have no further questions so far.

---

### Official Review · Reviewer_8ich · 2023-10-31

**Soundness:** 1 poor
**Presentation:** 1 poor
**Contribution:** 2 fair
**Rating:** 3
**Confidence:** 5

**Summary:**

The paper proposes a causal exploration strategy to leverage the causal structure of the data and model training to increase the sample effficency.

**Strengths:**

1. The problem addressed in the paper is very interesting.
2. The paper conducts experiments to demonstrate the effectiveness of the framework.

**Weaknesses:**

1. The paper is poorly written and mainly unorganized. The main contribution of the paper is completely lost.
2. Causal discovery from observational data is known to be unidentifiable with extra assumptions on the data generating process. The paper employs the PC algorithm which is also known to only be able to identify the causal graph up to CPDAG. The paper needs to discuss the unidentifiability issue and its effect on the performance of the proposed method.
3. The paper needs to provided theoretical and empirical analysis of the unidentifiability of causal discovery on the overall performance.

**Questions:**

Please refer to the weaknesses.

---

> ### Author Response · Authors · 2023-11-15
> **Thanks for the review**
>
> We greatly appreciate your insightful comments and time devoted to our work. We attempt to address all the concerns as follows.
>
> **Contribution**$\quad$We believe that our key contributions are summarized at the end of the introduction section, which are:
>
> * In order to enhance the sample efficiency and reliability of model training with causal knowledge, we introduce a novel concept: causal exploration, and focus particularly on the domain of task-agnostic reinforcement learning.
>
> * To efficiently learn and use structural constraints during world model learning, we design a novel weight-sharing-decomposition schema that can avoid additional computational burden.
>
> * We demonstrate the effectiveness of causal exploration across a range of demanding reinforcement learning environments.  Theoretically, we show that, given strong convexity and smoothness assumptions, our approach attains a superior convergence rate compared to non-causal methods. Empirically, experimental results on synthetic data demonstrate the ability to learn accurate world models with exceptional data utilization, surpassing existing exploration methods in complex scenarios. Notably, our approach also produces outstanding performance in challenging tasks such as traffic light control and MuJoCo, highlighting its practicality in real-world applications.
>
> Sorry for any inconvenience caused in your reading and we will make a better organization in the revised version.
>
> **Causal identifiability**$\quad$We sincerely appreciate your thoughtful comments. Consider that we focus on a Markov decision process (MDP) where the underlying states $\boldsymbol{s}_t$ are observed, the causal assumptions for our method are the Markov condition and faithfulness assumption. Besides, the identifiability theory of MDP has been established in prior works [1-3]. [2] also provides theoretical guarantees for the correctness of the causal graph we identified. We acknowledge the need for providing a theoretical analysis for this in the main paper and we will add this in the revised manuscript.
>
> [1] Biwei Huang, Fan Feng, Chaochao Lu, Sara Magliacane, and Kun Zhang. Adarl: What, where, and how to adapt in transfer reinforcement learning. arXiv preprint arXiv:2107.02729, 2021.}
>
> [2] Ding et al., “Generalizing goal-conditioned reinforcement learning with variational causal reasoning”, NeurIPS 2022.
>
> [3] Wang et al., “Causal dynamics learning for task-independent state abstraction”, ICML 2022.

---

> > ### Comment · Reviewer_8ich · 2023-11-22
> > **Thank you for your response!**
> >
> > I appreciate the authors' response. I have carefully read the rebuttal and other reviews. While some concerns have been addressed, I maintain that certain key issues raised by reviewers remain unresolved. My main concern is with respect to the paper's writing, contributions, and theoretical analysis. Nevertheless, I sincerely acknowledge and value the authors' dedication in addressing my concerns.

---

### Official Review · Reviewer_mVqv · 2023-11-02

**Soundness:** 3 good
**Presentation:** 3 good
**Contribution:** 2 fair
**Rating:** 5
**Confidence:** 3

**Summary:**

Building upon previous methods, this paper introduces an intrinsic reward to enhance performance. It analyzes the convergence rate and achieves promising experimental results.

**Strengths:**

Building upon previous methods, this paper introduces an intrinsic reward to enhance performance. It analyzes the convergence rate and achieves promising experimental results.

**Weaknesses:**

The paper's analysis of efficiency is currently insufficient, relying solely on experimental findings. However, the theoretical analysis of the paper is not enough to fully support the author's claims.

**Questions:**

* What does "t" in equation 5 signify? Does it refer to the optimization turn or the time step? If it aligns with the time step notation from the previous section, it might be challenging to understand why we should evaluate the difference between two errors at different time steps.

* Does the shared schema have a bad effect on the identifiability of causality?

* Which strategy can enhance efficiency? The shared schema appears to introduce an additional layer of parameters. Adding extra intrinsic rewards in equation 5 seems to increase computational costs.

* I appreciate the theoretical analysis of convergence, but this paper primarily focuses on efficiency, such as exploration efficiency and whether the convergence rate is better than what was previously required. The theoretical analysis and the point the paper aims to make appear somewhat disconnected.

* How is the dataset D_h collected? If the dataset is fixed and predetermined, learning parameters based on the collected dataset doesn't pose an exploration challenge. In Model-Based Reinforcement Learning (MBRL), exploration is intertwined with data collection.

---

> ### Author Response · Authors · 2023-11-15
> **Thanks for the review**
>
> We appreciate your thoughtful comments and time devoted to reviewing this paper, and hope the following response properly addresses your concerns.
>
> **Equation 5**$\quad$The method in [1] trains a strategy based on active reward to determine whether to annotate the incoming data for a supervised classifier. If the active reward value is positive, it indicates that the annotated data is helpful for the training process. Similarly, we use active rewards to encourage agents to explore toward data that is helpful for training. Therefore, active rewards are used to measure the training quality. To maintain consistency in the order of prediction loss, we need to ensure that the agent can receive prediction errors and active rewards at every time step $t$. That is, $t$ in equation 5 refers to the time step.
>
> [1] Meng Fang, Yuan Li, and Trevor Cohn. Learning how to active learn: A deep reinforcement learning approach. arXiv preprint arXiv:1708.02383, 2017.
>
> **Identifiability**$\quad$The identification of causal relationships is completed by the PC algorithm. In addition, the input of the world model only contains information about the parent nodes, so the shared schema does not affect causal identification.
>
> **Enhancing efficiency**$\quad$In our paper, "boosting efficiency" entails that agents can achieve comparable or superior performance with fewer sample size. As has been studied in [2,3,4], causal knowledge improves sample efficiency during training but calls for factorization. The shared schema reduces the computation burden caused by causal factorization. Besides, the scale of both the shard and decomposed layers can be regulated so there is no introduction of an additional layer even compared with non-causal methods. What's more, introducing active reward does indeed lead to an increase in computational complexity. But such a slight increase in computational complexity can be ignored compared to the improvement in model performance (Figure 9 in the supplementary materials). Furthermore, the additional computational costs can be reduced by controlling the scale of model $\phi$ and the test set $\mathcal {D}_h $.
>
> [2] Maximilian Seitzer, et al., Causal influence detection for improving efficiency in reinforcement learning. Advances in Neural Information Processing Systems, 34: 22905–22918, 2021.
>
> [3] Wang et al., “Causal dynamics learning for task-independent state abstraction”, ICML 2022.
>
> [4] Ding et al., “Generalizing goal-conditioned reinforcement learning with variational causal reasoning”, NeurIPS 2022.
>
> **Convergence analysis**$\quad$As is depicted in section B of the supplementary materials, "Denote $w_t^c$ as the network parameters taking causal constraints with respect to $w_t$ that dose not gather causal information. Theorem 3.1 shows that causal exploration gets a prediction error bound $\delta_t$ times lower at time $t$, where $\delta_t$ is a density measurement of causal matrix $D$." We believe this is consistent with what you said about “whether the convergence rate is better than what was previously required”.
>
> **Data collection**$\quad$Actually, we learn the parameters based on the collected data $B_t$ during exploration instead of $\mathcal{D}_h$ which is used for the calculation of active reward (line 9 and 12 in Algorithm 1).

---

### Official Review · Reviewer_SHAc · 2023-11-05

**Soundness:** 2 fair
**Presentation:** 2 fair
**Contribution:** 1 poor
**Rating:** 3
**Confidence:** 4

**Summary:**

This paper considers the problem of enhancing the data collection efficiency during the training the world model. To address this challenging issue, a new framework named causal exploration is proposed, to improve the performance of both causal discovery and model learning. During the exploration phase, the agent actively selects actions expected to yield causal insights most beneficial for world model training. This framework is built upon 2 key components: 1, an efficient online causal discovery method, and 2, a novel weight-sharing-decomposition schema to satisfy the causal constraint of the forward model.

**Strengths:**

This paper tried to address a very important question: how to utilize the underlying causal knowledge, and learn a world model in a task-agnostic fashion and data-efficient way? The proposed causal exploration is an interesting idea to address this question: it essentially allows "cold start" optimization of the world model with an unknown causal graph; and both the data collection policy ("i.e., experiments") and the causal discovery results are updated during the online learning process. The methodology is partially supported by convergence analysis. The experimental results on benchmarking different baselines could also be valuable to certain audiences in this field.

**Weaknesses:**

The main weaknesses of this paper are threefold:

- Lack of clarity in the presentation. Typically, for a paper like this would greatly benefit from a clearly written problem-setting section. However, this is currently missing and I can imagine that this will create major obstacles for understanding this paper, especially for researchers that are not from this field. These missing problem settings include but are not limited to:

  - the world model $f$ was never formally defined mathematically and is only briefly mentioned in Section 2.2 which is too late.
  - the authors mentioned the concept of "causal exploration" from the beginning of Section 2 without any mathematical description.
  - the goal of this paper is supposed to be "enhancing data collection efficiency to improve world model learning as well as causal discovery". However, this problem is not formally defined mathematically.

In general, I need to put all the pieces together and figure out them by myself. Although it did not take too much effort, the reading experience was not great either.


- Unclear causal assumptions and missing identification theory. The underlying causal assumptions of the proposed method were never discussed, and methods with different underlying causal assumptions were combined in an ad-hoc way. For example, as the authors might already be aware, the underlying factorization in Section 2.2, Eq 3 resembles the SCM models in causality literature. Then, why the authors did not consider applying causal discovery techniques that were specifically defined for SCM models instead of a generic PC algorithm? This should offer more advantages in terms of data efficiency since SCMs typically have stricter assumptions and stronger identification results. Moreover, the authors did not provide any theoretical analysis regarding the correctness of the identified causal graph, which is very crucial.


- Disconnection from the latest development of causal learning literature that addresses similar problems.
  -  The proposed causal exploration is closely related to a large field named active (Bayesian) causal discovery [3, 4, 5, 6], in which causal exploration can be viewed as a special case of active causal discovery, where the exploration policy exactly corresponds to the notion of experiment design of interventions.  In the current paper, all these related works are not properly acknowledged, benchmarked and compared.
  -  The computational complexity of Eq 3 is not a new problem, and the exact same weight-sharing schema has already been discussed in a number of literature [1, 2] in the context of causal deep learning.
  - Similarly, in the context of causal deep learning, efficient graph learning has already been discussed in a number of literature as well. These recent works since [1] mostly focused on differential graph learning which exploits the SCM assumptions as well as the corresponding identifiability theory. In contrast to the proposed efficient causal discovery that is based on discrete PC, I believe that the differential approach can be integrated into the causal exploration framework more easily. This is because the graph can be learned end-to-end and does not require a separate graph learning step.

- Flaws in experiment evaluation methods. If I understand correctly, the prediction errors in experiments are estimated on the test set used in exploration. This will give a significantly unfair advantage to the proposed method when compared with the baselines since the proposed method is designed to utilize such prediction error as a learning signal. The authors should consider using metrics such as the discrepancy between the learned model versus the ground true model (in synthetic examples) or its offline equivalence (in real-world scenarios).


Given the aforementioned weakness, I am not convinced by the significance and novelty of the contribution. However, I am open to discussions and if my concerns are addressed during the rebuttal, I will raise my score accordingly.


References:

[1] Zheng X, Aragam B, Ravikumar P K, et al. Dags with no tears: Continuous optimization for structure learning[J]. Advances in neural information processing systems, 2018, 31.

[2] Geffner T, Antoran J, Foster A, et al. Deep end-to-end causal inference[J]. arXiv preprint arXiv:2202.02195, 2022.

[3] Murphy K P. Active learning of causal Bayes net structure[R]. technical report, UC Berkeley, 2001.

[4] Tong S, Koller D. Active learning for structure in Bayesian networks[C]//International joint conference on artificial intelligence. Lawrence Erlbaum Associates ltd, 2001, 17(1): 863-869.

[5] Tigas P, Annadani Y, Jesson A, et al. Interventions, where and how? experimental design for causal models at scale[J]. Advances in Neural Information Processing Systems, 2022, 35: 24130-24143.

[6] Ivanova D R, Jennings J, Rainforth T, et al. CO-BED: Information-Theoretic Contextual Optimization via Bayesian Experimental Design[J]. arXiv preprint arXiv:2302.14015, 2023.

**Questions:**

See the Weakness section.

---

> ### Author Response · Authors · 2023-11-15
> **Thanks for the review (part 1)**
>
> Thanks a lot for your thoughtful comments and suggestions. Please see below for our responses to your specific comments.
>
> **Lack of problem-setting section**$\quad$Sorry for any inconvenience caused in your reading and we will add a Preliminary section before section 2 following your invaluable suggestions.
>
> **Causal assumptions and identification theory**$\quad$We sincerely appreciate your thoughtful comments. Consider that we focus on a Markov decision process (MDP) where the underlying states $\boldsymbol{s}_t$ are observed, the causal assumptions for our method are the Markov condition and faithfulness assumption. Besides, the identifiability theory of MDP has been established in prior works [7]. [8] provides theoretical guarantees for the correctness of the causal graph we identified. We acknowledge the need for providing a theoretical analysis for this in the main paper and we will add this in the revised manuscript. Moreover, we would like to clarify that the PC algorithm is also a common method for causal discovery in SCM models through the application of conditional independence tests, as demonstrated in [8,9]. Furthermore, [8] has proven that the use of independence tests for causal discovery tends to outperform and exhibit greater stability compared to other causal techniques like score-based methods for SCM in our settings. In addition, the ASR method precisely employs an end-to-end approach for embedding causal structure and world model learning. We have included it as a baseline method for experimental comparison (Figure 5).
>
> [7] Biwei Huang, Fan Feng, Chaochao Lu, Sara Magliacane, and Kun Zhang. Adarl: What, where, and how to adapt in transfer reinforcement learning. arXiv preprint arXiv:2107.02729, 2021.
>
> [8] Ding et al., “Generalizing goal-conditioned reinforcement learning with variational causal reasoning”, NeurIPS 2022.
>
> [9] Wang et al., “Causal dynamics learning for task-independent state abstraction”, ICML 2022.
>
> **Disconnection from SOTA**
>
> (1)We appreciate the reviewer's attention to this specific point. We would like to clarify that while the exploration policy in our work shares a conceptual similarity with the notion of experiment design in active causal discovery, the practical implications and challenges are different in the context of reinforcement learning. In traditional active causal discovery, experiment design often involves interventions to perturb the system and observe the causal effects. However, in reinforcement learning, interventions are typically challenging due to the sequential, interactive nature of decision-making. Agents in reinforcement learning environments operate in an ongoing interaction with the environment, making controlled interventions more complex compared to static observational settings. Our exploration policy, while conceptually related to experiment design, is designed to efficiently collect data for model learning rather than explicit interventions. This adaptation is necessary to accommodate the unique challenges posed by sequential decision-making in reinforcement learning settings. We will highlight this distinction more explicitly in the revised manuscript.
>
> (2)We greatly appreciate your provided references. However, after carefully reviewing them, we did not find a similar weight-sharing schema as employed in our work. In [1], the causal discovery is formulated as a continuous optimization problem, and the authors use the Lagrangian method for solving. This approach differs from ours, as we employ deep neural network structures. Additionally, [1] focuses solely on linear Structural Equation Models (SEMs), while our deep network has the capacity to approximate a wide range of linear and nonlinear functions. In [2], the authors merge the decomposed networks into a unified one by learning a training embedding $\boldsymbol{u}_i$ to serve as index information for each dimension, i.e., $f_i(\cdot)=f(\boldsymbol{u}_i,\cdot)$. However, it's crucial to note that $\boldsymbol{u}_i$ is present in the form of an input, whereas our structure involves passing inputs through a shared layer before transitioning to respective decomposed networks. Moreover, $\boldsymbol{u}_i$ is a d-dimensional vector in [2], whereas our shared network's layer count is dynamically adjustable.
>
> (3)Similar to the answer for causal assumptions, [8] has provided both theoretical and experimental evidence supporting the superiority and greater stability of using independence tests for causal discovery compared to other causal techniques, such as score-based methods for Structural Causal Models (SCM) in our settings. Furthermore, the ASR method precisely employs an end-to-end approach for embedding causal structure and world model learning. We have included it as a baseline method for experimental comparison, as shown in Figure 5.

---

> ### Author Response · Authors · 2023-11-15
> **Thanks for the review (part 2)**
>
> **Experiment evaluation methods**$\quad$We appreciate your valuable insights. But we would like to clarify that the prediction error serves as a suitable metric for assessing model training efficacy as is highlighted in [10] and has been used in several works like [8,9]. Its application to exploration data precisely reflects that our model achieves shorter training times and utilizes fewer samples. Furthermore, all baseline methods, with the exception of CID, employ intrinsic rewards for training. Points of distinction include: firstly, our intrinsic reward formulation encompasses active reward, which cannot be considered a specialized design for using prediction loss. Secondly, we provide additional causal structural information to the model. Moreover, CID employs a strategy of random sampling for model training, whereas utilizing intrinsic reward forms a more appropriate exploration approach. Certainly, we also utilize the discrepancy between the learned model and the ground truth model as an evaluation criterion. In contrast to a direct model-to-model comparison, we choose to compare the learned causal graph with the ground truth one (Figure 5). Additionally, the downstream performance for traffic signal control and the MuJoCo task, as depicted in Table 1 and Figure 7(b), adheres to the concept of offline equivalence in real-world scenarios as you mentioned.
>
> [10] Ke, Nan Rosemary et al. "Systematic evaluation of causal discovery in visual model-based reinforcement learning." arXiv preprint arXiv:2107.00848 (2021).

---

### Meta-Review · Area_Chair_mgyw · 2023-12-06

**Metareview:**

The paper proposes a causal exploration scheme to improve the sample efficiency of learning dynamics models in task-agnostic reinforcement learning problems.  The empirical results of the scheme are promising. However, all of the reviewers agree that the paper is below the bar for publication because (1) it lacks a good articulation of its contributions (because of a sub-par related work section that is missing crucially relevant papers), and (2) it should theoretically analyze causal identifiability questions of their proposed problem setting.
During the author-reviewer discussion, some of the empirical evaluation concerns raised by the reviewers were clarified by the authors but the points related to (1) and (2) remain.

**Justification For Why Not Higher Score:**

The paper can be substantially stronger if it references the rich literature on causal dynamics learning in RL and clarifies the novelty and significance of its contributions w.r.t existing methods that learn the causal relationships and dynamics directly. It will help to include these related methods as baselines in the experiments as a more representative example of the state of the art.

**Justification For Why Not Lower Score:**

N/A

---

### Decision · Program_Chairs · 2024-01-16

Reject